# The Importance of Sharing Data in Systems Biology

**DOI:** 10.3390/metabo13010099

**Published:** 2023-01-07

**Authors:** Elisha M. Wood-Charlson

**Affiliations:** Environmental Genomics and Systems Biology Division, E.O. Lawrence Berkeley National Laboratory, Berkeley, CA 94720, USA; elishawc@lbl.gov

**Keywords:** systems biology, FAIR data, open science

## Abstract

Systems biology research spans a range of biological scales and science domains, and often requires a collaborative effort to collect and share data so that integration is possible. However, sharing data effectively is a challenging task that requires effort and alignment between collaborative partners, as well as coordination between organizations, repositories, and journals. As a community of systems biology researchers, we must get better at efficiently sharing data, and ensuring that shared data comes with the recognition and citations it deserves.

## 1. Introduction

Advancements in science are becoming more frequent, but there are still many questions that remain unanswered. Many of these pressing questions/challenges are large and complex, requiring researchers to approach them from many perspectives. The field of “systems biology” has long sought a holistic approach to research, and is therefore well-positioned to continue to expand in scope and impact. One origin for systems biology was proposed as a merger between molecular biology, bolstered by advances in genomics and high-throughput technologies, and nonequilibrium thermodynamics, which provided the basis for understanding biochemical pathways and networks [1]. While this framing focused on the cellular level, systems biology includes work across scales—molecular, cellular, organismal, communities, and all that within an environmental context. Technical advancements have enabled this increase in scope, and the impact of the questions being answered span biological, ecosystem, and medical sciences, as well as newer fields such as synthetic biology, and even reach into bioengineering [2]. Many of the articles in this Special Issue provide illustrative examples of the breath, depth, and scope of systems biology and its ability to connect and advance knowledge. The aim of this short commentary is to highlight the importance of sharing data for systems biology approaches to be effective and impactful. Without data that are sharable, comparable, and reproducible (among other things), the large complex science questions will remain challenging.

## 2. Your Data Are Someone Else’s Metadata

Science is a collaborative venture. Data produced by one person are best served when built upon by others. However, when data are collected, they are often viewed from a domain-specific perspective. For example, the activity of a particular gene in a given environment may indicate metabolic activity, but in order to understand what that means, there must be sufficient metadata to provide context: what environment, what type of cell(s), what were the experimental conditions (temperature, light, nutrients, etc.), and so forth. This metadata might actually be primary data for collaborators on the project, e.g., tracking chemical profiles across gradients. Complex questions often require the integration of many types of data (and metadata), many types of analytical tools to explore those data/metadata, and many perspectives to understand and make conclusions from those data/metadata. This level of data integration requires data to be findable, accessible, interoperable, and reusable—otherwise known as the FAIR Principles [3]. Although making data FAIR is just the beginning, it is a critical first step. We recently published “10 simple rules on getting and giving credit for data”, and Rule 1 summarizes the FAIR principles [4]. In brief, data should be findable through a unique persistent identifier (e.g., DOI), accessible via a standard protocol (e.g., HTTPS), stored in interoperable file/data format(s), and have enough provenance and information about prior use to be reusable. The FAIR principles have become a shared mission across most funders, data repositories, publishers, libraries, etc., and are more likely to be rewarded by proper citation in manuscripts. So, if you need help making your data FAIR, please ask.

## 3. Coming Together to Make Data Comparable and Reproducible

As mentioned above, the concept of FAIR data is just the foundation. Systems biology requires the integration of diverse data types, often collected using different methods, each with a different context, and described by different metadata. These components are often important for assessing comparability and ensuring reproducibility. Implied in the FAIR principles is the need for open science, and standardized ways to capture and share this information. What those standards are is often not explicit, as standards are—like their data—often domain-specific. As a community, it is critical that we agree on common standards, and adopt them in our everyday data reporting. This ensures that data are readable and comprehensible by humans as well as machines. Additionally, as data and metadata are often made available digitally, we are ever more reliant on search engines to help us find and access data.

Yes, these extra layers take extra effort. However, think about the last time you tried to find and use someone else’s data. Typically, that experience also takes a lot of effort—effort by someone that is not familiar with the data. When interviewed on topics around sharing data and data reuse, many researchers (>75%) expressed concern about the misinterpretation of data, or data being used in ways it was not originally intended [5]. Researchers also tend to have more confidence in using data collected by others when there are details about collection, standards, and provenance [5]. Persistent identifiers (PIDs) are critical to FAIR data and can be assigned to all research products, including physical samples (IGSN—International GeoSample Number), digital outputs, including data and protocols (DOIs—digital object identifiers), and even the people (ORCID—Open Researcher and Contributor ID) and organizations (ROR—Research Organization Registry) conducting the research. For an example of how PIDs can be applied to biological data, namely a physical sample collected from the environment (e.g., soil or water), processed in the laboratory (e.g., DNA sequencing), and analyzed using bioinformatic tools, please see Table 2 in the work of Wood-Charlson et al. [4].

Many standards already exist, and there are programs that provide training to help researchers learn how to use them effectively. For systems biology, depending on where you are, check out Elixir Europe [6], the Australia Biocommons [7], H3ABioNet [8] in Africa, and in the United States—the National Institute of Health (NIH) Office of Data Science Strategy [9]. More information about these groups and their “Life Science Data Infrastructures” efforts can be found through the Research Data Alliance (RDA [10]). There are also FAIR data initiatives and open science training available globally through GO-FAIR [11], OpenAIRE [12], and CODATA, the Committee on Data International Science Council [13]. As a community of systems biologists that need diverse data sets to address complex questions, making our data FAIR and open, and including standardized information that make data comparable and reproducible just makes sense.

## 4. Benefits of Making Data FAIR, Comparable, and Reproducible

In addition to making your data open and easier for others to discover and reuse correctly (and for you to do the same with theirs), another benefit of making data FAIR is that it is easier for people to cite your data. Tenopir [5] reported that 92.1% of researchers in their survey felt that having their data cited was important. We wrote “10 simple rules on getting and giving credit for data” [4] to help make this easier, both for data producers as well as data consumers. The main actionable components for linking diverse data across publications are: (1) assigning FAIR data persistent identifiers (typically DOIs, the same as a publication) and (2) including the DOI in the data availability statement and the manuscript reference section alongside other references. If you have a lot of data sets, combined and recombined, consider turning them into a new data set collection and getting your own data set DOI. The DOI citation chain (your new DOI still cites the original DOIs) captures the full provenance, and some data analysis and sharing platforms (like the DOE Systems Biology Knowledgebase [14]) will track, cite, and help you publish these collection data sets, as well as your original data.

Another benefit of making data FAIR also means getting ahead of data sharing mandates. If you have not already been instructed to share your data, you likely will be soon. Many governments, funding agencies, foundations, and publishers have announced, and often already implemented, data sharing mandates. Example data mandates include: Australia’s National Health and Medical Research Council (NHMRC) data sharing policy—20 September 2022; NIH Data Management and Sharing policy—25 January 2023; and the United States White House Office of Science and Technology Policy (for all US federally funded scientific data)—31 December 2025. 

## 5. Conclusions

Best practices for sharing data take effort, and mandates do not always result in adherence to these [15]. We have to make it easier. We have to take the pain out of data sharing [16]. The FAIR principles are a starting point, but it will take a community effort to align on standards and best practices around documenting and sharing metadata, as well as the data they describe. There continue to be new and improved ways to capture research outputs from complex projects through RAIDs (Research Activity Identifier) [17] or RO-Crate [18]. Additionally, given the applied and varied nature of systems biology questions, we could benefit from lessons learned in clinical research, where system-level reviews are also important and evaluated by the Preferred Reporting Items for Systematic reviews and Meta-Analysis (PRISMA) checklist [19]. Resources are out there. And, since we do not have to invent them, we could do more to modify and adopt them. We have some pressing science questions to solve.

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
