# Peer review of "The Importance of Sharing Data in Systems Biology"

_metabolites, 2023, doi:10.3390/metabo13010099_

Round 1

Reviewer 1 Report

Dear authors, 

- Introduction is too short

- Abstract is not a correct framework and needs to improve

- Conclusion part must be improved

- lack of enough references is abvious

Author Response

  • Introduction is too short
    • Without additional information on what might be missing from the introduction to make the article more clear, I prefer to not add text just to make it longer.
  • Abstract is not a correct framework and needs to improve
    • As this is not a classic research article, I am not sure what “correct framework” I should be adhering to? I have focused on feedback from other reviewers to improve the abstract.
  • Conclusion part must be improved
    • Again, without additional information on what could be improved, I am unsure how to best proceed. I focused on feedback from the other reviewers. Hopefully that is sufficient.
  • Lack of enough references is obvious
    • Additional references have been added.

Reviewer 2 Report

The author  provides an excellent commentary on systems biology in lieu of FAIR guidelines

The author whilst mentioning  Australia Biocommons and H3ABioNet could subtly delve on Asia specific resources as well, for example what Ensembl Asia has been doing. The best practices in these niche areas could be discussed in a sentence or two.

Liekwise, a word on PRISMA guidelines would be a nice addition

A pictorial abstratc would bring life to the commentary

Line 46:  FAIR principles coud be abbreviated on first hand 

Scores on a scale of 0-5 with 5 being the best 

Language: 4

Novelty; 4

Brevity: 4

Scope ans relevance: 4.5

Author Response

  • The author provides an excellent commentary on systems biology in lieu of FAIR guidelines
    • Thank you!
  • The author whilst mentioning Australia Biocommons and H3ABioNet could subtly delve on Asia specific resources as well, for example what Ensembl Asia has been doing. The best practices in these niche areas could be discussed in a sentence or two.
    • I wasn’t able to find anything specific to Asia via Ensembl (still run mostly out of Europe, for Europe) or the Research Data Alliance, which is a global community. Found international training opportunities by CODATA, which are active in Asia, so added that link.
  • Likewise, a word on PRISMA guidelines would be a nice addition
    • Great recommendation, added to the conclusion alongside a few other items.
  • A pictorial abstract would bring life to the commentary
    • My artistic ability is very basic, but I will provide something hopefully illustrative. Thank you.
  • Line 46: FAIR principles could be abbreviated on first hand 
    • Thanks for this recommendation. I prefer to spell out FAIR at first mention in a manuscript.

Reviewer 3 Report

I suggest to accept this article as it is because there are a lot useful of information in system biology research area.

Author Response

  • I suggest to accept this article as it is because there are a lot useful of information in system biology research area.
    • I am very glad you found it useful!

Reviewer 4 Report

Comments to the manuscript “The Importance of Sharing Data in Systems Biology”, To begin with, this is a very interesting comment to point out the importance and relevance to share data for the scientific community in the biological field, specifically to use it in several topics to make them comparable and reproducible, or also for modelling or to analyze complex topics through meta-analysis studies.  

Comments to the author

In the Abstract, to mention that it would be important that journals should have their own science domains (repositories) to save data bases related with the published article and with a specific format (ordered).

Line 46, why the word FAIR is capitalized? It is an acronym, or you mean the word fair.  

Line 124, change “authors” by “author”, in singular form instead of plural.

In the conclusion title, change 5 instead of 6.

Conclusions should not have citations, please move citation 14 and 15 to another part of the text.

Author Response

  • In the Abstract, to mention that it would be important that journals should have their own science domains (repositories) to save data bases related with the published article and with a specific format (ordered).
    • I feel this would be too onerous on journals, could increase already very high publishing costs, and would further split data across even more locations, further complicating comparability and reuse. Many journals endorse/recommend repositories for their authors to consider. I added a statement to emphasize that coordination includes organizations, repositories, and journals. Thank you for that idea/recommendation.
  • Line 46, why the word FAIR is capitalized? It is an acronym, or you mean the word fair.
    • Added capitals to the words to indicate it was an acronym.
  • Line 124, change “authors” by “author”, in singular form instead of plural.
    • Fixed
  • In the conclusion title, change 5 instead of 6.
    • Fixed
  • Conclusions should not have citations, please move citation 14 and 15 to another part of the text.
    • Other review articles in Metabolites appear to have citations throughout the conclusions section, so this does not appear to be common practice for this journal. I will leave them and defer to the editor.

Reviewer 5 Report

The author has elegantly provided a commentary on the importance of sharing the data in systems biology. I have a minor suggestion.

It would be helpful to include a a short summary of the Reference no.4 by Wood-Charson et. al, as it is referred at multiple places. Especially referring to Table 2 and 10 rules from Ref. 4 would make the flow of readability a bit difficult. 

Author Response

  • It would be helpful to include a short summary of the Reference no.4 by Wood-Charlson et. al, as it is referred at multiple places. Especially referring to Table 2 and 10 rules from Ref. 4 would make the flow of readability a bit difficult. 
    • Thank you for that feedback. Additional context has been added.

Round 2

Reviewer 1 Report

This version of paper is acceptable.